# Women with Schizophrenia over the Life Span: Health Promotion, Treatment and Outcomes

**DOI:** 10.3390/ijerph17155594

**Published:** 2020-08-03

**Authors:** Alexandre González-Rodríguez, Armand Guàrdia, Aida Álvarez Pedrero, Maria Betriu, Jesús Cobo, Sidharta Acebillo, José Antonio Monreal, Mary V. Seeman, Diego Palao, Javier Labad

**Affiliations:** 1Department of Mental Health, Parc Taulí University Hospital, Institut d’Investigació i Innovació Parc Taulí (I3PT), Autonomous University of Barcelona (UAB), 08280 Sabadell, Barcelona, Spain; aalvarezp@tauli.cat (A.Á.P.); jcobo@tauli.cat (J.C.); sacebillo@tauli.cat (S.A.); jmonreal@tauli.cat (J.A.M.); dpalao@tauli.cat (D.P.); jlabad@tauli.cat (J.L.); 2Department of Mental Health, Parc Taulí University Hospital, 08280 Sabadell, Barcelona, Spain; aguardia@tauli.cat (A.G.); mbetriu@tauli.cat (M.B.); 3Department of Psychiatry, University of Toronto, Toronto, ON M5P 3L6, Canada; mary.seeman@utoronto.ca

**Keywords:** women, schizophrenia, psychosis, gender, antipsychotics

## Abstract

Women with schizophrenia show sex-specific health needs that differ according to stage of life. The aim of this narrative review is to resolve important questions concerning the treatment of women with schizophrenia at different periods of their life—paying special attention to reproductive and post-reproductive stages. Review results suggest that menstrual cycle-dependent treatments may be a useful option for many women and that recommendations re contraceptive options need always to be part of care provision. The pregnancy and the postpartum periods—while constituting vulnerable time periods for the mother—require special attention to antipsychotic effects on the fetus and neonate. Menopause and aging are further vulnerable times, with extra challenges posed by associated health risks. Pregnancy complications, neurodevelopmental difficulties of offspring, cancer risk and cognitive defects are indirect results of the interplay of hormones and antipsychotic treatment of women over the course of the lifespan. The literature recommends that health promotion strategies need to be directed at lifestyle modifications, prevention of medical comorbidities and increased psychosocial support. Careful monitoring of pharmacological treatment has been shown to be critical during periods of hormonal transition. Not only does treatment of women with schizophrenia often need to be different than that of their male peers, but it also needs to vary over the course of life.

## 1. Introduction: Health Care for Women with Schizophrenia

Schizophrenia is a severe mental disorder with a lifetime prevalence of nearly 1% [1]. The World Health Organization has tried, through its global burden of disease (GBD) studies, to quantify the burden of disease attributable to schizophrenia. The most recent data are from GBD 2016 [1]. These studies have concluded that schizophrenia is a disease of low prevalence, exhibiting little sex difference in prevalence, but resulting in substantial burden of disease for both sexes [2].

One important cause of burden is the medical comorbidity found in individuals with schizophrenia [1,3]. Researchers agree that this high level of ill health is partly caused by ill-advised behavior such as excessive smoking. It is well known, for instance, that cardiovascular disease, the main cause of death in schizophrenia, is strongly associated with tobacco use [3]. Although behaviors such as smoking are considered modifiable factors, circumstances associated with schizophrenia—poverty, lack of employment, stress, absence of social/emotional support, mental illness stigma, significant cognitive deficiencies and the cumulative effects of antipsychotic medication, taken together, pose often insurmountable barriers to change [4].

There have been many investigations into gender differences in schizophrenia. Men are reported to show more premorbid difficulties, but the overall prevalence of schizophrenia symptoms is similar in the two sexes [5]. With respect to comorbidity, men and women are vulnerable to somewhat different health problems [6]. Recent studies have highlighted problems that are specific to women, namely women’s sexual functioning, contraceptive practices and reproductive issues [7]. One concern is the underuse of contraception in the schizophrenia population, an issue that mental health providers frequently fail to address. When patients with psychotic illness were confined to mental hospitals, sexual activity and the potential for pregnancy did not arise. When first generation antipsychotic medication became available and patients were treated in the community, the high doses used very often were enough to prevent conception [7]. This is no longer true. Because a majority of male patients with schizophrenia exhibit negative symptoms that render them asocial and unmotivated, it was wrongly assumed that neither men nor women were interested in sexual relationships. This is also not true. Unless women with schizophrenia are married or partnered, contraception is still not commonly discussed in most therapeutic conversations.

Clinical practice guidelines for schizophrenia rarely identify sex-specific requirements [8] although attention is being paid in the more recent literature to sexually dimorphic features of drug efficacy and tolerability [9]. Hormonal effects on drugs have led to a more general appreciation of hormonal influence on the dimorphic expression of schizophrenia in women and men [10]. For instance, recent studies have found that levels of follicle-stimulating hormone (FSH), dehydroepiandrosterone, and luteinizing hormone (LH) are closely associated with the severity of schizophrenia symptoms [10].

### Aims

The aim of this review is to illustrate the different treatment needs of women with schizophrenia during different periods of their life (reproductive years, perinatal periods, postmenopausal years). To achieve this objective, several questions regarding women with schizophrenia will be addressed:(a)What aspects of schizophrenia differ in men and women?(b)What features of sexual and reproductive life of women with schizophrenia require special treatment?(c)How effective and how safe are psychotropic medications during the perinatal period? On what does efficacy and safety during this period depend?(d)What changes to antipsychotic treatment need to be made at the postmenopausal stage of life in women with schizophrenia? What other health needs deserve attention at this age?

## 2. Methods

To obtain answers to these questions, we carried out a non-systematic, narrative, critical review of the literature on the topic of women and schizophrenia. We focused on answers to our four questions rather than attempting to provide an all-encompassing, systematic review. We searched the PubMed and Google Scholar databases for English and Spanish papers published in the last decade that referenced women, schizophrenia and health. Additionally, we expanded our search to include a few frequently cited, older papers. A digital search was combined with a manual check of references from all studies in the field of women and schizophrenia. Restrictions were not applied with regard to study design. Where many papers cited the same earlier work, the earlier publication was the one included. Whenever there were multiple publications by the same research team on a topic, the earliest paper was preferentially cited.

The following search terms were used: women AND schizophrenia. Studies were included if they met the following eligibility criteria: (a) studies focusing exclusively on women with schizophrenia, (b) investigations referring to the needs of women with schizophrenia as they pertained to their sexual and reproductive life or (c) studies focused on the safety and efficacy of psychotropic medications during perinatal and postmenopausal periods.

All authors scanned abstracts and titles from 500 initial potentially relevant papers, most of which were excluded because they did not contribute new information. By consensus, we selected 79 articles that provided new light on the questions of interest. All selected papers helped to address at least one of the four main questions we were aiming to resolve.

## 3. Results

### 3.1. Gender Differences in Schizophrenia

In comparison with men, women present for treatment at a later age, generally respond more robustly to antipsychotic treatment and require hospitalization less often over the course of illness [11]. As in the larger population, women attempt suicide more often, but completed suicide rates are lower than men’s [12]. Altogether, women appear to enjoy a better quality of life, show fewer negative symptoms and exhibit less social and clinical disability than men [13]. Nevertheless, in terms of physical health, their comorbidity risk is relatively higher [14].

Researchers in this area agree that gender differences can be partially explained by neuromodulators, namely, female sex hormones. The estrogen protection hypothesis, first formulated at the beginning of the 1980s, posits that the brain action of estrogens that affects mood, cognition and behavior is responsible for women’s comparative advantages over men with respect to schizophrenia prognosis [15]. Estrogen exerts a protective effect against the progress of several neuropsychiatric disorders, not only schizophrenia [16]. Recent work demonstrates that, in women with schizophrenia, symptoms are more severe, relapses occur more readily, and hospital admissions increase when the concentration of circulating sex hormones is reduced, for instance, during the late luteal and early follicular phase of the menstrual cycle, immediately postpartum and following menopause. Likewise, symptoms lessen when hormone levels rise, namely during the mid-luteal stage of the menstrual cycle and over the course of pregnancy [17]. Some studies have found that premorbid estrogen levels in women who later develop schizophrenia are already lower than they are in the general population [14], which further implicates estrogen in the pathogenesis of this disease.

Female hormones also impact drug distribution, metabolism and elimination. High levels of estrogen, by reducing levels of glycoprotein, raise the free concentration of antipsychotic medications, thus increasing their entry into the brain and allowing them to reach more target receptors. Moreover, the presence of estrogen at the dopamine receptor site helps to slow the transmission of dopamine, an excess of which is thought to lead to psychotic symptoms. Estrogen also regulates the activity of specific cytochrome P450 (CYP) enzymes that metabolize drugs, thus potentially increasing the levels of at least some antipsychotic drugs in women. This is most evident for clozapine and olanzapine, both metabolized by CYP1A2 [9].

With regard to antipsychotic drugs in general, it is important to remember that they are all lipophilic, e.g., they are stored in fatty tissue. Women’s bodies are composed of more adipose tissue than men’s so more drug accumulation occurs in women [18]. A significant amount of evidence supports the notion that there are strong sex differences in amount of adipose tissue and, by implication, sex differences in metabolism [18]. This means that long-acting depot antipsychotics need not be administered to women as frequently as to men. For the same reason, drug discontinuation will induce relapse of symptoms far quicker in men than it does in women. The storage of lipophilic drugs in fatty tissue also accounts for adverse effects such as weight gain, more prevalent in women than in men. For this reason, adipose tissue is now being targeted by developers of antipsychotic drugs [19].

Another pharmacokinetic sex difference is the slower elimination rate in women than in men because of a less abundant hepatic and renal blood flow, and a slower glomerular filtration rate [9]. During the luteal phase of the menstrual cycle, women’s gastric acid secretion is reduced, and gastrointestinal transit time takes longer, which slows absorption of drugs. Another important consideration is that women with schizophrenia, more often than men, are prescribed drugs in addition to antipsychotics (antidepressants, antimanic agents, contraceptives, analgesics...) eliciting potential drug interactions that can modify the availability of the antipsychotics. Finally, female sex hormones bring about changes in liver microsomal oxidation activity that also affects antipsychotic bioavailability [17].

Turning to adverse effects of antipsychotic medication, women are more likely than men to respond with greater weight gain, more metabolic symptoms, more sexual dysfunctions and more cardiovascular disease [20,21]. Some risk factors are almost exclusive to women, namely Torsade de Pointes and hypercoagulability states (venous thromboembolism, pulmonary embolism and cerebrovascular events, including stroke). The use of oral contraceptives and hormone replacements are added risk factors for the latter complications [22]. Hyperprolactinemia is more prevalent in women because baseline levels are already higher than they are in men [20,21]. High prolactin disrupts menstrual cycles, diminishes fertility and leads to galactorrhea, amenorrhea, hirsutism and acne [19]. It is an important cause of osteoporosis and has been suspected of contributing to breast cancer risk [23]. Because of gender differences in immunity, the agranulocytosis induced by clozapine is thought to be more frequent in women than in men [24].

Adjusting drug doses according to the ovulatory cycle and reproductive phase is not currently practiced, but the literature suggests that it perhaps should be. There is some evidence for boosting the dose of antipsychotic medication during periods of lower estrogenic concentration. Alternatively, estradiol or selective estrogen receptor modulators can be added to the drug regimen to reduce symptoms and perhaps also to permit a reduction of antipsychotic dose and, in this way, alleviate antipsychotic adverse effects [24].

In summary, clinicians need to keep in mind that menstrual cycle and reproductive stages in women require special consideration when prescribing antipsychotic medication (Table 1).

### 3.2. Sexual and Reproductive Health in Premenopausal Women with Schizophrenia

Most women with schizophrenia seen in psychiatric practice are of reproductive age [24,25] and their lives are being impacted by both the illness and its treatment. Menses, sexual relationships, birth control measures, marriage, pregnancy, labor and delivery, breastfeeding and child rearing are all affected. One example, described by Miller and Finnerty [26] as well as by Sethuraman et al. [27], is that women with this diagnosis are insufficiently informed about the use of contraception. Many women with schizophrenia are sexually active, usually outside of a stable partnered relationship, with the result that there is a 24.3–47.5% prevalence of unwanted pregnancy in this population [7]. Another study found the prevalence to be 39.4%, with alcohol use substantially increasing the risks [28]. Because women with schizophrenia may be unreliable about taking a daily contraceptive pill, other contraceptive methods such as intrauterine devices, depot injections of progesterone and tubal ligation, may be the preferred choice [26]. Care providers should also be prepared to discuss morning after pills. Mental health professionals can refer these women to gynecologists who are up to date on contraceptive methods. When sexual encounters are casual, there is also the danger of sexually transmitted disease [6,29], about which frank discussions are essential. Women with schizophrenia are socially isolated and often badly informed about health risks.

Unwanted pregnancy necessitates decisions about abortion or relinquishing the newborn to adoption, decisions that are stressful and have long-lasting, potentially deleterious effects on women [30,31]. There is also the risk of sexual exploitation in this population [32], which requires appropriate counselling and, sometimes, legal intervention.

The use of hormonal contraception may pose problems in this population. The levels of hormone in oral contraceptives are decreased by antiepileptic mood stabilizers [33], which can be part of the treatment regimen in women with schizoaffective disorder. A low hormone level may not prevent pregnancy. Antiepileptics taken during pregnancy raise the risk of teratogenesis [34]. There is another risk to oral contraceptives. They decrease the concentration of certain psychotropic drugs and can, thus, result in psychiatric decompensation (Table 2) [35,36].

Psychotic symptoms vary across the menstrual month, depending, to an extent, on estradiol levels [37]. Lande and Karamchandani [38] have reported that admissions to psychiatric units are more common in the late luteal and early follicular menstrual phase and lowest in the high estradiol ovulatory phase. Symptom severity, according to some reports, also rises during the late luteal [37,38].

Whether under treatment with antipsychotics or not, menstrual dysfunction is common in this population because of frequently high testosterone and low estrogen blood concentrations [39,40]. Co-morbidity with polycystic ovary syndrome is relatively frequent [41,42]. The hyperprolactinemia induced by antipsychotic medication further decreases estrogen levels and leads to menstrual irregularities such as amenorrhea, which is associated with low fertility [37,43]. Low levels of estrogen can theoretically contribute to the cognitive difficulties that are associated with schizophrenia because estrogens promote neurotrophic synthesis in the hippocampus and prefrontal region, areas that sustain memory and judgement. Irregular menstrual cycles predict defects in psychomotor speed, verbal fluency and verbal memory (Table 3) [44].

Reproductive health includes precautions against domestic abuse, often prevalent in this population, especially during pregnancy [45]. Mothers with schizophrenia also require support [46] and preventive measures against loss of child custody [47]. These include treatment adherence, awareness of signs of impending relapse, advance preparation for hospitalization or other crisis, familiarity with legal issues pertaining to mental health and custody, attendance at parenting courses and ready availability of parental surrogates.

### 3.3. Treatment of Schizophrenia during the Perinatal Period

Effective treatment during the perinatal period is vital in women with a previous history of psychotic disorders [48,49]. Although all medications are best avoided during pregnancy in order to protect the fetus, in the case of serious maternal illness such as schizophrenia, it is vital that mother’s health remain stable. This means continuing antipsychotic medications, although dosage can be adjusted and kept as low as possible [49]. In the postpartum period, women are particularly vulnerable to psychotic relapse so that antipsychotic medication is essential at that time, although it may interfere with breastfeeding [50]. Mental health screening and support for perinatal women is a necessity for all women [51]. The postnatal period is the right time to help with planning effective ongoing contraception [52].

Obstetric outcomes are poorer for women with severe mental illness when compared to the general population [53], due to poverty, comorbidities, smoking, alcohol use, gaps in prenatal care and lack of social support. There is an iatrogenic component to the risk. Antipsychotic treatment contributes to obstetric complications and to complications for the newborn, such as stillbirth, neonatal death, infant developmental difficulties [53], but the conclusion that these are caused by antipsychotic medication is not supported by sufficient replicated evidence.

Coughlin and collaborators [54] included 13 cohort studies in their systematic review, with 6289 antipsychotic-exposed women and 1,618,039 relatively unexposed women. Antipsychotic exposure during pregnancy was associated with an increased risk of major malformations, heart defects, preterm delivery, small-for-gestational-age births, elective termination of the pregnancy and decreased infant birth weight at delivery. It was not associated with risk of large-for-gestational-age births, stillbirth or spontaneous abortion [54]. There were no associations with specific antipsychotics. The most controversial issue—and the one with least evidence—is the possibility that antipsychotics negatively influence long-term development of the child [55]. Important to remember is that untreated schizophrenia during pregnancy is itself an independent risk factor, not only for congenital malformations, but also for more general health risks affecting mother and fetus and infant [56,57].

Recent studies have focused on the relative risk for gestational diabetes mellitus (GDM) in severe mental disorders. Panchaud and collaborators [58], in a sample from the Massachusetts General Hospital national pregnancy registry for atypical antipsychotics, assessed 303 women without previous diabetes who were exposed to second generation antipsychotics before pregnancy. Of these women, 10.9% presented with GDM (10.7% in the non-exposed sample). After adjustment for putative confounding factors (maternal age, race, marital status, employment status, level of education, smoking and primary psychiatric diagnosis,) the OR was only 0.79 (0.40–1.56). By contrast, in a retrospective study of 539 pregnant women with mental disorders, Galbally and collaborators [59] found a significantly elevated risk for GDM (20.9%) in pregnant women with psychotic disorders, compared with women with non-psychotic but still severe mental illnesses and a nearly threefold risk if compared to the expected population rate (8.3%). Some specific antipsychotic agents, such as risperidone, clozapine or high-dose quetiapine were associated with a four-fold increased risk for GDM after adjusting for maternal age and body mass index. In this sample, common confounding factors, such as tobacco smoking, alcohol consumption or illicit drug use, were not associated with an elevated risk for GDM [59].

Ellfolk and collaborators [60] included 1,181,090 pregnant women and their singleton births in a population-based birth cohort study using national register data extracted from the Drugs and Pregnancy database in Finland. Pregnant users of second generation antipsychotics were shown to have an increased risk for GDM, for cesarean section, for infants born large for gestational age and for preterm birth. Infants in the drug group were also more likely than controls to suffer from neonatal complications. The number and severity of neonatal complications did not differ between first and second generation antipsychotic users [60].

Preventive strategies for complications in the obstetric care of mothers with schizophrenia should ideally address three relevant factors: lifestyle modifications, increase psychosocial support and prevention of medical comorbidities [61].

Should antipsychotic be discontinued or decreased during pregnancy? Results of a recent systematic review showed how abrupt discontinuation of antipsychotics in mothers with bipolar disorder or schizophrenia led to symptomatic relapses during pregnancy [55]. The authors concluded that, “after taking into account the parents will and after they provide informed consent, the most reasonable and less harmful choice for treating future mothers with bipolar disorder or schizophrenia appears to be maintaining them at the safest minimum dosage” [55].

With respect to the risk of relapse during the postpartum period, the magnitude of risk may differ according to several sociodemographic and clinical factors [62]. Universally, however, the production of estrogens abruptly declines [63]. The balance among interrelated hormones, drug metabolism and other pharmacokinetic factors all change in the postnatal period [64]. Psychotic relapses can be prevented by ensuring the availability of psychiatric care, which may include home visits, lactation advice, insomnia advice and treatment and the provision of home help and psychosocial support. Psychoeducation about illness and child care is also needed [65]. It is very difficult for new mothers to attend psychiatric appointments—telepsychiatry contacts or home visits need to be arranged [66].

In summary, recommended management of women with schizophrenia during pregnancy and the perinatal period should include discussion of the risks and benefits of antipsychotic treatment and the fact that dosages may vary as hormone levels change [67]. Treatment during this period needs to be individualized [68]. Close liaison between psychiatry and obstetrics will ensure adherence to prenatal directives, important for the prevention of obesity, diabetes and potential complications of pregnancy. Ensuring adequate housing and psychosocial support for pregnant and postpartum women with schizophrenia is a critical element of care [69,70].

### 3.4. Treatment and Health Care Needs in Women with Schizophrenia Postmenopause

By definition, menopause is the end of ovulation, accompanied by the loss of the neuroprotection conferred by estrogens during the reproductive period [23,71]. The decline in sexual hormones is associated with a worsening of psychotic symptoms and increased side effects of antipsychotic treatment [23]. The literature agrees that women with schizophrenia after menopause require raised doses of antipsychotics in order to stay well. The advent of menopause may bring extra psychological and medical problems that necessitate assessment, preventive measures, as well as attention to diet and exercise and the prevention of loneliness and isolation [24].

There is good evidence for the observation that the loss of estrogens exerts negative impact on psychotic symptoms. A study carried out by González-Rodríguez and co-workers [72] found that antipsychotic response depends on the duration of time that has passed since menopause onset. In other words, time since menopause was found to be negatively associated with good response to antipsychotic medications. The same group investigated the role played in this association by the follicle stimulating hormone/luteinizing hormone (FSH/LH) ratio, but the results were inconclusive [73]. A recent study supports the identification of hormonal phenotypes among women with schizophrenia, suggesting that it may help to improve existing treatments and implement personalized-medicine strategies that address the heterogeneity seen in schizophrenia [10].

Hormone therapies, with their benefits and risks, have been proposed as adjunctive treatments for postmenopausal women with schizophrenia [74]. Estrogen receptor modulators, such as raloxifene, have been effectively added to standard treatment in order to reduce symptom severity [75]. Reduced severity of symptoms is important because antipsychotics, especially in older age, increase the risk for metabolic, cardiovascular and neurologic adverse effects [23]. Such treatment may, however, add to the risk of venous thrombophlebitis and embolism [23,76]. On the other hand, the use of hormone adjuvants may permit lowering of antipsychotic dose while still maintaining symptom control.

Advancing age brings with it increased comorbid medical illness that is not always effectively treated in the schizophrenia population [77]. Sufficient preventive measures may not be taken. Lindamer and colleagues [77] found that women with schizophrenia took part in pelvic examinations, Pap tests and mammography screening significantly less frequently than controls [77]. This relatively low participation in cancer screening is also true for breast and colon screening [78,79].

The menopausal and postmenopausal age is a vulnerable period for women with schizophrenia. This is when vulnerability to psychosis increases, response to antipsychotic medication decreases and medical problems, some of them secondary to antipsychotic treatment, increase in frequency. Clinicians need to pay special attention to the health needs of women with schizophrenia as they age.

## 4. Discussion

Women with schizophrenia show specific health needs at different stages of their life—all of which require evaluation and clinical management. This narrative review provides several answers to the four questions asked at the beginning.

What aspects of schizophrenia differ between men and women? Extrapolating from the results of our search, the answer to this question is: specific symptom, relationships, hormonal transitions, maternal/fetal conflicts, comorbidity and antipsychotic response. These are critical issues that need to be addressed during treatment.

What features of sexual and reproductive life of women with schizophrenia require special emphasis? Our search found effective contraception to be important. While mental health professionals agree that family planning should be provided to this population of women, recent evidence attests to the fact that only 25% of professionals raise this topic with their patients. Concerns of women related to family planning are important. For women who are not ready to become parents, it is important to thoroughly discuss issues of contraception. Counseling to protect against sexual and domestic exploitation should be also provided. Clinicians need to be aware that pregnancy, the postpartum period and the menopausal years are all periods of high vulnerability for women with schizophrenia. Extra precautions need to be taken.

How effective and how safe are psychotropic medications during the perinatal period? Upon what does efficacy and safety during this period depend? The consensus is that stopping effective medication during pregnancy, and especially immediately after delivery is a dangerous practice because it places women with schizophrenia at risk for serious relapse. On the other hand, the literature shows that antipsychotic treatment during this period is not without risk. The recommendation is to balance risks and benefits by calibrating antipsychotic dose to individual needs.

What changes to antipsychotic treatments need to be made at the postmenopausal stage of life in women with schizophrenia? What other health needs deserve attention at this age? Investigators suggest that, when estrogen levels fall, psychotic symptoms increase, so that antipsychotic doses need to be raised. There is a realization, however, that this may expose women to adverse effects with negative medical consequences. The adjunctive use of hormone replacement or treatment with estrogen receptor modulators appears very useful at this time.

## 5. Conclusions

The literature suggests that woman-specific treatment include menstrual cycle-dependent antipsychotic dosing, attention to drug interactions, plus intervention, as needed, in cases of sexual exploitation or domestic abuse, unwanted pregnancy and child custody decisions. Parental support can make a critical difference in the lives of women and their children. At the time of menopause, recommendations include attention to adequate antipsychotic dosing, prevention of adverse effects and screening for health risks.

Optimal interventions in women with schizophrenia differ according to stage of life. At all stages, preventive strategies need to address risks and benefits of treatment, lifestyle modifications, prevention of medical comorbidities and the mobilization of psychosocial support.

This review has several limitations and strengths that should be noted. The first limitation is that it is a narrative review rather than a systematic review. It is a non-systematic, narrative, critical review of the literature on the topic of women and schizophrenia rather than an attempt to provide an all-encompassing review of the literature. This method has allowed us to include classic papers in the field that have helped to answer the four questions specified at the beginning.

## Figures and Tables

**Table 1 ijerph-17-05594-t001:** Hormonal and metabolic characteristics of reproductive stage women with schizophrenia.

Low testosterone and high estrogen concentration
High proportion of adipose tissue
Reduced liver and kidney blood flow
Exposure to concomitant treatments
Disturbance in hepatic microsome oxidation
Reduced gastric acid secretion during luteal phase of cycle
Long gastrointestinal transit time during luteal phase of cycle

**Table 2 ijerph-17-05594-t002:** Women with schizophrenia—problems related to sex.

Uncommitted sexual relationships
Lack of contraception
Sexual exploitation
Sexually transmitted diseases
Unwanted pregnancies

**Table 3 ijerph-17-05594-t003:** Menstrual dysfunctions and reproductive difficulties found in women with schizophrenia.

Amenorrhea
Low fertility
Psychosis exacerbations during luteal stage of menses
Polycystic ovaries

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
