# Peer review of "Women with Schizophrenia over the Life Span: Health Promotion, Treatment and Outcomes"

_ijerph, 2020, doi:10.3390/ijerph17155594_

Round 1

Reviewer 1 Report

Dear authors,

A novel and interesting contribution that deals with a current topic. This document shows the results of a data analysis in specialized literature, but does not provide details on the review protocol used, the eligibility criteria of the analyzed documents or the methods used to retrieve the data, so that it can be replicated. As suggested by the magazine, the guidelines of the PRISMA Declaration must be fully followed. An in-depth review is recommended.

Kind Regards. 

Author Response

Reviewer 1

1.”A novel and interesting contribution that deals with a current topic. This document shows the results of a data analysis in specialized literature, but does not provide details on the review protocol used, the eligibility criteria of the analyzed documents or the methods used to retrieve the data, so that it can be replicated. As suggested by the magazine, the guidelines of the PRISMA Declaration must be fully followed. An in-depth review is recommended.”

We entirely agree with Reviewer 1 that a methods description is needed in our review. We added a methods section in order to describe the methodology we followed to reach our results. We have now clarified that our review was a non-systematic and narrative review with the main aim of addressing some research questions. We clarified search terms, databases used and which criteria we have used.

This review is an in-depth, but not systematic, approach to specific research questions. It is an in-depth review as it guides strategy and supports evidence-based decision making within the topic of women and schizophrenia.

As the main aim of this review was to summarize current and recent work on schizophrenia and women in a narrative design, we did not carry out a systematic review as per the PRISMA statement. This narrative and critical review allows us to review several aspects of reproductive and post-reproductive topics in schizophrenia, and to propose health promotion strategies.

Reviewer 2 Report

The manuscript by Gonzalez-Rodriguez et al. discuss an important and little investigated topic, namely the gender differences in schizophrenia development and the role of the sex hormones fluctuation over the women lifetime in the disease progression. The subject is of a high importance for the clinical practice. Effective treatment of schizophrenia, especially in the critical periods such as pregnancy, postpartum, lactation period, and menopause is crucial for patient’s quality of life. The authors make a detailed review of the studies in the field. However, several clarifications are required.

Major remarks:

Paragraph 3 - Could authors discuss in more details the role of the hormonal contraception as a hormonal therapy and discuss the intrauterine coil as a contraceptive method for women diagnosed with schizophrenia

Conclusion -The manuscript will have a higher value if authors come up with specific and more detailed suggestions for interventions during critical periods such as pregnancy and menopause and strategies for prevention of unwanted pregnancy, sexual exploitation, domestic abuse, obstetric complications, and child custody.

Minor remarks:

Abstract, line 18 and introduction, line 59 require clarification – menstrual cycle, pregnancy, lactation and menopause are stages in lifetime, not stages of reproductive and post-reproductive disease, neither different stages of schizophrenia

Paragraph 3 – please explain the reasons why women with schizophrenia are not well informed about the use of contraception and elaborate in the conclusion how it could be prevent.

Table 2 and 3 Authors could consider to change the title to “Sexual difficulties” (for table 2) and “Menstrual dysfunctions and reproductive difficulties” for table 3. The Amenorrhea and Galactorrhea then, should be state in table 3. It is not clear if cognitive difficulties are directly dependent and are a result from hormonal levels during menstrual cycle.

The numbers of the paragraphs in the manuscript should be corrected.

Author Response

Reviewer 2

Major remarks:

1-“Paragraph 3 - Could authors discuss in more details the role of the hormonal contraception as a hormonal therapy and discuss the intrauterine coil as a contraceptive method for women diagnosed with schizophrenia”

We have added further discussion on the different methods of contraception for women with schizophrenia. We have emphasized that a variety of appropriate methods of contraception for women who suffer from schizophrenia should be carefully explored and that methods such as intrauterine devices, progesterone depot injections or tubal ligation for women who do not wish to expand their families should be considered.

2-“Conclusion -The manuscript will have a higher value if authors come up with specific and more detailed suggestions for interventions during critical periods such as pregnancy and menopause and strategies for prevention of unwanted pregnancy, sexual exploitation, domestic abuse, obstetric complications, and child custody.”

We have expanded suggestions for prevention strategies in the perinatal period and the menopause to include some potential interventions for complications that may arise at these critical periods of the life-span.

Minor remarks:

1-“Abstract, line 18 and introduction, line 59 require clarification – menstrual cycle, pregnancy, lactation and menopause are stages in lifetime, not stages of reproductive and post-reproductive disease, neither different stages of schizophrenia”

We have clarified this question. We have modified these lines and all references within the body of the manuscript to better explain that we were referring to stages in a woman’s life: reproductive and menopausal stages.

2-“Paragraph 3 – please explain the reasons why women with schizophrenia are not well informed about the use of contraception and elaborate in the conclusion how it could be prevent.”

Thank you for your suggestion. We have added the reasons why women with schizophrenia have been under-informed about contraceptive methods. We have also recommended prevention strategies in the third paragraph of the conclusions.

3-“Table 2 and 3 Authors could consider to change the title to “Sexual difficulties” (for table 2) and “Menstrual dysfunctions and reproductive difficulties” for table 3. The Amenorrhea and Galactorrhea then, should be state in table 3. It is not clear if cognitive difficulties are directly dependent and are a result from hormonal levels during menstrual cycle.”

We have corrected the titles of Tables 2 and 3 according to the recommendations from Reviewer 2.

We have also included amenorrhea and galactorrhea in Table 3 and removed cognitive difficulties from this table. Amenorrhea and galactorrhea were eliminated from Table 2, as they are not related to sexual difficulties.

4-“The numbers of the paragraphs in the manuscript should be corrected.”

We have renumbered the paragraphs in the manuscript. Furthermore, we have included a new methods section and a results section where we included all the results subsections.

Reviewer 3 Report

The research topic is definitely great to the field.

However, in the content, it is not readiness for the publication. For example, the structure of the paper is not in a fine form, e.g. the numbering is wrongly displaying; and model, and research methodologies are not easily identified. 

Also it lacks of the discussion part of the paper.  

Author Response

Reviewer 3

1-“The research topic is definitely great to the field. However, in the content, it is not readiness for the publication. For example, the structure of the paper is not in a fine form, e.g. the numbering is wrongly displaying; and model, and research methodologies are not easily identified. “

We entirely agree with Reviewer 3 that the numbering of the paper was wrong. We have re-numbered all the sections and subsections according to the recommendation. We added a methods section, and a results section where we detailed several subsections according to the results we found.

We have better defined our goals, and the questions we aimed to address when conducting this narrative non-systematic review. Research methodology was further detailed in the “Methods section”.

2-“Also it lacks of the discussion part of the paper “

We have expanded the conclusions section and added discussion of the results of the paper as they pertain to the four main questions we address.

Round 2

Reviewer 1 Report

In this new version, the authors have added a new section where they describe the methodology followed to reach the results.

They have expanded the conclusions section and included the main limitations of the study. I believe that this version can be published.

Thank you for responding to the reviewer's comments.

Kind Regards